# A Hessian View of Grokking in Mathematical Reasoning

Zhenshuo Zhang[†]     Jerry W. Liu[‡]     Christopher Ré[‡]     Hongyang R. Zhang[†]

[‡]Department of Computer Science, Stanford University, Stanford, CA
[†]Khoury College of Computer Sciences, Northeastern University, Boston, MA

## Abstract

Mathematical reasoning is a central problem in developing more intelligent language models. An intriguing phenomenon observed in mathematical arithmetics is grokking, where the training loss of a transformer model stays near zero for an extended period until the validation loss finally reduces to near zero. In this work, we approach this phenomenon through a view of the Hessian of the loss surface. The Hessian relates to the generalization properties of neural networks as it can capture geometric properties of the loss surface, such as the sharpness of local minima. We begin by noting in our experiments that high weight decay is essential for grokking to occur in several arithmetic tasks (trained with a GPT-2 style transformer model). However, we also find that the training loss is highly unstable and exhibits strong oscillations. To address this issue, we consider adding regularization to the Hessian by injecting isotropic Gaussian noise to the weights of the transformer network, and find that this combination of high weight decay and Hessian regularization can smooth out the training loss during grokking. We also find that this approach can accelerate the grokking stage compared to existing methods by at least $50\%$ measured on seven arithmetic tasks. Finally, to understand the precise cause of grokking, we consider a Hessian-based measurement for multi-layer networks and find that this measure yields non-vacuous estimates of the generalization errors observed in practice. We hope these empirical findings can facilitate future research towards understanding grokking (and generalization) in mathematical reasoning.

## 1   Introduction

Mathematical reasoning in large neural networks [1] is a central issue in the design of more intelligent, interactive language models, especially in scenarios that require precise, step-by-step logical operations. An intriguing phenomenon that has been observed in arithmetic tasks is grokking [19], where a transformer model exhibits a delayed yet sudden generalization of training data even as the training curve has converged. In this paper, we analyze grokking behavior in arithmetic tasks by examining the Hessian of the transformer model's loss surface.

Power et al. [19] demonstrate that for a range of modular arithmetic tasks, grokking can occur on two-layer transformers, whereby the training loss remains near zero for a long period until the validation loss also converges to zero. However, the training loss can exhibit dramatic oscillations. To motivate this work, we begin by applying regularization methods to SGD and examine their effect on grokking. First, we find that high weight decay is crucial for grokking in arithmetic tasks. In other words, we find that the validation accuracy does not increase to near perfect when weight decay is moderate (Fig. 1a and 1b). Moreover, even after adding high decay, the training curve still exhibits notable variations, leading to unstable training (with training accuracy reduced to zero; see Fig. 1c).

The 4th Workshop on Mathematical Reasoning and AI at NeurIPS 2024 (MATH-AI 24).

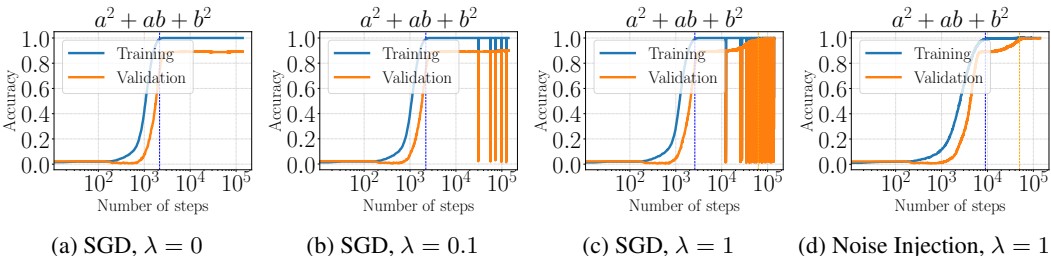

Figure 1: Training behavior with different weight decay (denoted as $\lambda$) and Hessian regularization.

To address these issues, we consider regularization methods that regularize the Hessian. These methods are known to improve generalization by reducing the sharpness of solutions in the loss surface [6, 22]. In particular, we consider regularizing the Hessian by first adding noise to the weights of the transformer before computing its gradient. We observe that this noise injection provides an approximately unbiased estimate of the trace of the Hessian. Surprisingly, we find that, along with high weight decay, this noise injection algorithm can now smooth out the oscillations in the training curve. Moreover, we also find that this Hessian regularization can reduce the number of grokking steps during training by at least 28%, measured on three arithmetic tasks.

Finally, to understand these results, we develop a preliminary theoretical analysis. We examine a Hessian-based generalization measure motivated by PAC-Bayes analysis [17, 11, 2]. We find that by measuring a Hessian-vector product on the weight space, we can provide a non-vacuous estimate of the generalization errors. Note that the phenomenon of delayed generalization is known since classical works on boosting [3]. Our contribution is to provide a Hessian view of this phenomenon since the Hessian can be measured from data.

In summary, we find that by using high weight decay and noise injection to regularize the Hessian, we can effectively reduce the instability that has commonly been observed for training transformers on arithmetic tasks. Second, this combined regularization can further reduce the number of grokking steps. Third, we develop a Hessian-based measurement that can give a non-vacuous estimate of the generalization error. We hope these findings can facilitate future research on understanding grokking in the mathematical reasoning of large models.

## 2    Regularization of Loss Surface Hessian

Previous works have indicated that grokking requires an appropriate choice of weight decay [18, 14]. However, using weight decay alone can still lead to oscillations of the training curve. To address this issue, we explore an alternative approach, where we regularize the loss Hessian matrix, which can provide more fine-grained control on the loss surface, such as sharpness. To instantiate the Hessian regularization, we add a random noise variable to the weight matrices of a transformer network. In particular, let $\ell(f_W(x), y)$ denote the loss of a neural net $f_W$ (parameterized by $W$), given an input pair $(x, y)$. Let $U$ be a random sample from an isotropic Gaussian (with the same dimension as $W$), whose variance has been scaled by $\sigma^2$. We consider the following noise injection update:

$$W \leftarrow W - \frac{\eta}{2}\left(\nabla\ell\left(f_{W+U}(x), y\right) + \nabla\ell\left(f_{W-U}(x), y\right)\right), \tag{1}$$

for some learning rate $\eta$. In particular, $\sigma^2$ determines the level of regularization in this procedure. To see that this update regularizes the Hessian, we notice that equation (1) is equivalent to applying SGD to the stochastic optimization objective of

$$\mathbb{E}_U\left[\ell(f_{W+U}(x), y)\right] \approx \ell(f_W(x), y) + \frac{\sigma^2}{2}\nabla^2\ell(f_W(x), y) + O(\sigma^3).$$

In practice, we add the noise injection along both the positive and negative directions of $U$. This helps eliminate the variance that appears from the first-order Taylor's expansion term above [22].

# 3 Experimental Results

## 3.1 Results on arithmetic tasks

Our experimental setup follows the work of Power et al. [19]. We focus on evaluating the grokking phenomena with arithmetic tasks, which correspond to equations of the form $(a \circ b) \bmod p = c$. "$a$," "$\circ$," "$b$," "$=$," and "$c$" are separated tokens, where "$c$" is the prediction goal. For each task, we generate $a \in [p]$ and $b \in [p]$, resulting in a total of $p^2$ (in our settings, $p = 97$) unique data. Specifically, we select $a^2 + ab + b^2 = c$ to illustrate our findings, and more experiments of different equations are shown in Appendix A. We compare our regularization method to naive SGD and SAM [6]. SAM is based on a constrained minimax optimization formulation that penalizes the worst-case perturbations.

**#1: Stabilizing the training curves.** We observed that all the approaches can induce grokking in our experiments, as shown in Figure 2. After the training loss converges, we observe a sudden increase in validation accuracy to over 99%. Although all these methods can exhibit stable training before convergence, we find that during the grokking phase, SGD experiences sharp fluctuations in the training curve, with training accuracy dropping close to zero, which also caused the validation accuracy to drop to nearly zero. By contrast, both SAM and our noise-injection method can maintain stable training loss values during the grokking phase, which avoids dramatic fluctuations.

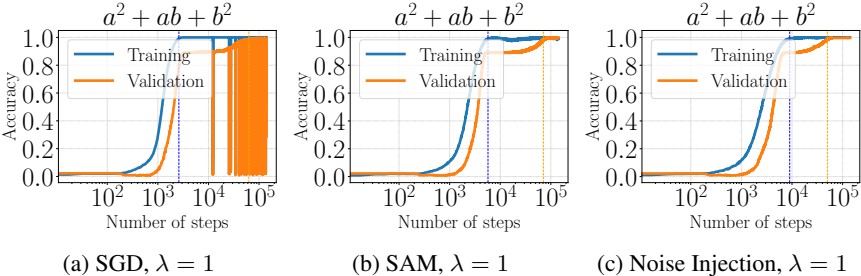

(a) SGD, $\lambda = 1$  (b) SAM, $\lambda = 1$  (c) Noise Injection, $\lambda = 1$

Figure 2: Illustrating the grokking phenomenon of SGD, SAM, and Hessian regularization.

**#2: Reducing the steps of grokking steps.** We also report the comparison of the number of training steps from the point where the training accuracy has converged to near 100% to the point where validation accuracy convergences to near 100%. We report the results for different approaches in Table 1. We observe that our approach requires fewer steps than SGD. In some tasks, our approach doesn't even need grokking steps to generalize. We also note that SAM, which penalizes the largest eigenvalue of the Hessian, requires more steps than noise injection.

Table 1: Number of grokking steps observed for different methods.

|      | $a+b$ | $a \times b$ | $a/b$ | $a^2+b^2$ | $a^2+ab+b^2$ | $a^2+ab+b^2+a$ | $a^3+ab$ |
|------|-------|--------------|-------|-----------|--------------|----------------|----------|
| SGD  | 36480 | 5040         | 89868 | 7890      | 48280        | 150263         | 83776    |
| SAM  | 30240 | 3620         | 44718 | 9170      | 26452        | 220116         | 0        |
| Ours | 14826 | 0            | 24576 | 3700      | 0            | 51187          | 0        |

## 3.2 Results on algorithmic tasks

We also evaluate our findings on the Needle-in-a-Haystack task, following the setting of Zhong et al. [23]. Specifically, we have an input sequence $[m_1, c_1, m_2, c_2, ..., m_k, c_k, m_u]$, where $m_i$ are different markers and $c_i$ are corresponding values. The last element is a marker $m_u, u \in [1, k]$ which indicates the goal marker. The model is trained to learn to search for the marker in the previous sequence and give the corresponding value $c_u$.

More surprisingly, we observe that the grokking phenomenon does not occur when using SGD, although the training curve also experiences slight fluctuations, and the validation accuracy remains low. The number of grokking steps of SAM and noise injection are 12288 and 5632, respectively: see Figure 3.

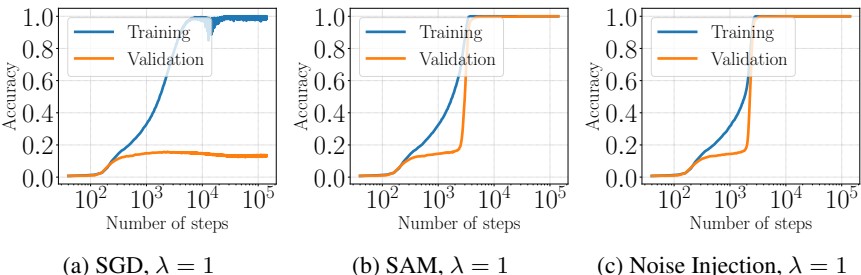

(a) SGD, $\lambda = 1$        (b) SAM, $\lambda = 1$        (c) Noise Injection, $\lambda = 1$

Figure 3: Illustrating the grokking phenomenon of SGD, SAM, and Hessian regularization.

## 4   Nonvacuous Generalization Error Estimates with Hessian

Toward rigorously understanding the above empirical results, we consider the PAC-Bayes analysis framework. In particular, we consider a linear PAC-Bayes bound [16, 2], which holds with probability $1 - \delta$ for any $\delta > 0$:

$$L_{\mathcal{Q}}(f_W) \leq \frac{1}{\beta}\hat{L}_{\mathcal{Q}}(f_W) + \frac{C\Big(KL(\mathcal{Q}||\mathcal{P}) + \log\frac{1}{\delta}\Big)}{2\beta(1-\beta)n}, \text{ for any } \beta \in (0,1). \tag{2}$$

Above, $\mathcal{Q}$ is the posterior hypothesis distribution of the learning algorithm. $\mathcal{P}$ is the prior distribution of the learning algorithm. $C > 0$ is an upper bound on the loss value. $L$ and $\hat{L}$ refer to the expected and empirical risks. For example, in the context of fine-tuning foundation models, one may view $\mathcal{P}$ as the weight of the pretrained model (plus some small perturbations), and $\mathcal{Q}$ is the fine-tuned model weight [11]. For a complete statement, see Lemma B.2 in Appendix B.

**Derivation of a Hessian-based measure:** Let $U \sim \mathcal{Q}$ be a random variable drawn from a posterior distribution $\mathcal{Q}$. We are interested in the perturbed loss, $\ell_{\mathcal{Q}}(f_U(x), y)$, which is the expectation of $\ell(f_U(x), y)$ over $U$. Using Taylor's expansion, we get that

$$\ell_{\mathcal{Q}}(f_W(x), y) - \ell(f_W(x), y) \leq \left(\sum_{i=1}^{L}\left\langle\Sigma_i, \nabla_i^2[\ell(f_W(x), y)]\right\rangle + C_1\|\Sigma\|_F^{3/2}\right), \tag{3}$$

where $\Sigma_i$ is the population covariance matrix of the perturbation added to layer $i$, and $\nabla_i^2$ is the Hessian matrix with respect to the weights at layer $i$ of $f_W$. See Lemma B.1 in Appendix B for the complete statement of this result.

Based on equation (3), next, we apply the PAC-Bayes bound from equation (2) to an $L$-layer transformer neural network $f_W$ parameterized by $W$. We note that the KL divergence between the prior and posterior distributions, which are both Gaussian, is equal to $\sum_{i=1}^{L}\left\langle\Sigma_i^{-1}, v_i v_i^{\top}\right\rangle$, where $v_i$ is the distance between the initialized weight and the trained weight at layer $i$.

Next, it remains to minimize the sum of the Hessian estimate, and the above KL divergence in the PAC-Bayes bound will lead to a different covariance matrix for every layer. Let $\nabla_i^{2^+}$ denote the truncated Hessian matrix where we set the negative eigenvalues of $\nabla_i^2$ to zero. We have that

$$\sum_{i=1}^{L}\left(\left\langle\Sigma_i, \nabla_i^2[\ell(f_W(x), y)]\right\rangle + \frac{1}{n}\left\langle\Sigma_i^{-1}, v_i v_i^{\top}\right\rangle\right)$$

$$\leq \sum_{i=1}^{L}\left(\left\langle\Sigma_i, \nabla_i^{2^+}[\ell(f_W(x), y)]\right\rangle + \frac{1}{n}\left\langle\Sigma_i^{-1}, v_i v_i^{\top}\right\rangle\right). \tag{4}$$

By applying equations (4) and (3) back to equation (2), and minimizing over $\beta$, we will derive an upper bound on the generalization error (between $L(f_W)$ and $\hat{L}(f_W)$) that is equal to:

$$\alpha := \max_{(x,y)\in\mathcal{D}}\sum_{i=1}^{L}\frac{\sqrt{v_i^{\top}\nabla_i^{2^+}\ell(f_W(x), y)v_i}}{\sqrt{n}}, \tag{5}$$

where $n$ is the size of the sample set and $\mathcal{D}$ is the unknown distribution where the samples are drawn.

Having introduced the Hessian measure, we now report the results from measuring the above $\alpha$ in the grokking experiments and compare $\alpha$ with the empirically observed generalization errors. The results are shown in Figure 4 below. We can see that $\alpha$ now gives a nonvacuous upper bound on the generalization error. Importantly, while this can also be achieved with standard methods such as $k$-fold cross-validation, the Hessian can reveal more structures (e.g., sharpness) of loss surfaces.

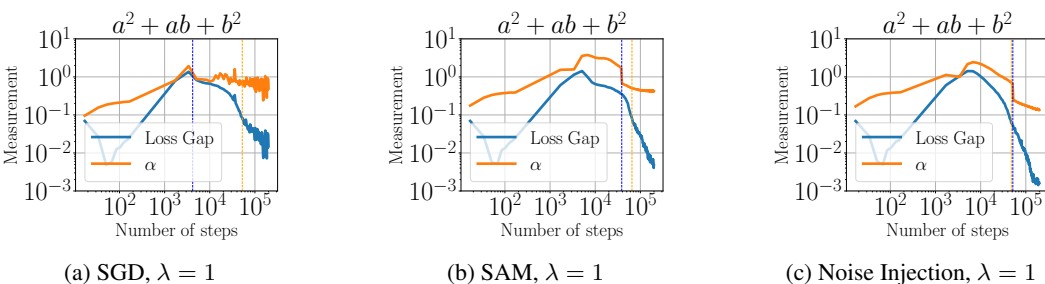

(a) SGD, $\lambda = 1$        (b) SAM, $\lambda = 1$        (c) Noise Injection, $\lambda = 1$

Figure 4: The Hessian measurement $\alpha$ correlates with the empirically observed generalization errors for training neural networks while grokking.

## 5 Related Work

**Grokking.** The grokking phenomenon, first proposed by Power et al. [19], illustrates that with continued training over several epochs, the validation loss eventually decreases and converges after training loss does not decrease further after converging. Extending the study of grokking, Liu et al. [15] conducted experiments across diverse datasets, including images, language, and graphs, expanding the area of grokking. Previous research predominantly focused on how training configurations influence grokking. Davies et al. [5] explored the relationship between grokking and double descent concerning pattern learning. Huang et al. [9] examined the impact of model and dataset sizes on grokking. Nanda et al. [18] highlighted the critical role of weight decay in grokking, noting that insufficient weight decay prolongs the process of grokking. Thilak et al. [20] linked grokking to the slingshot mechanism, interpreting it as a form of implicit regularization. More recently, Lee et al. [14] introduced Grokfast, a method designed to accelerate grokking by amplifying the gradients' low-frequency components. Theoretical investigations of why grokking occurs have recently been studied [21]. In particular, Xu et al. [21] provably demonstrate grokking in two-layer ReLU networks trained by gradient descent on XOR cluster data where a constant fraction of the training labels are flipped.

**Hessian and optimization algorithms.** Historical studies on second-order methods for training multi-layer networks primarily focus on optimization methods like Newton or quasi-Newton and employ the Hessian matrix to adjust learning rates [13, 12, 4]. Although they estimate the spectral information of the Hessian by computing Hessian-vector products, they do not explore the dynamics of the Hessian throughout training. In the Neural Tangent Kernel (NTK) analysis [10], the Hessian matrix is treated as a random features matrix, which remains fixed during training. The estimation of spectral density through Stochastic Lanczos Quadrature is discussed by Ghorbani et al. [7]. Additionally, Grosse et al. [8] have investigated scaling up influence functions in large neural networks, which includes innovative techniques for computing Hessian-inverse vector products.

## 6 Conclusion

This paper explored the issue of generalization in arithmetic tasks. We analyze the grokking phenomenon through a view of the Hessian matrix of the loss surface. We find that using a high weight decay and noise injection can smooth out the oscillations commonly observed in SGD training of arithmetic tasks. Another benefit of this regularization is that we could accelerate the grokking stage, reducing the number of training steps required for model generalization. Finally, we find that a Hessian-based measurement can give a nonvacuous estimate of the generalization errors in various modular arithmetic tasks.

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

# A  Experiment Details

## A.1  Tasks

**Modular Arithmetic**  We consider the following list of modulo tasks where $p = 97$:

- $a + b \,(mod\, p) = c$,
- $a \times b \,(mod\, p) = c$,
- $a/b \,(mod\, p) = c$,
- $a^2 + b^2 \,(mod\, p) = c$,
- $a^2 + ab + b^2 \,(mod\, p) = c$,
- $a^2 + ab + b^2 + a \,(mod\, p) = c$,
- $a^3 + ab \,(mod\, p) = c$.

**Needle-in-a-Haystack**  This task assesses model performance on long input sequences. The input consists of a sequence $[m_1, c_1, m_2, c_2, ..., m_k, c_k, m_u]$, where $m_1, \ldots, m_k$ are distinct markers with corresponding values $c_1, \ldots, c_k$. The final marker $m_u$ requires the model to locate its prior occurrence and output the associated value $c_u$. In our task, following [23], our sequences contain between 1 and 30 markers, and we uniformly select each $m_i \in \{1, \ldots, 127\}$ and $c_i \in \{128, \ldots, 158\}$.

## A.2  Parameters

We include the parameters we use to define our modular arithmetic and needle-in-a-haystack tasks below.

**Optimizers**  For all the methods, we use a weight decay of $\lambda = 1$, a learning rate equal to $10^{-4}$, and a maximum number of epochs of $1.4 \times 10^5$, with batch size 512. For SAM, we set $\rho = 0.05$. For our noise-injection method, we set $\sigma = 0.01$.

**Model**  For all modular arithmetic tasks, we use a model dimension of 128, whereas for the needle-in-a-haystack task, we use a model dimension of 256. For all modular arithmetic tasks except $a^3 + ab$, we use a 1-layer model. For $a^3 + ab$ and needle-in-a-haystack tasks, we use a 2-layer model instead. We use 4 attention heads for all experiments.

| Tasks | Training ratio | Batch size | Learning rate | Weight decay | Layers | Model dimension | Attention heads |
|---|---|---|---|---|---|---|---|
| $a + b$ | 0.3 | 512 | 1e-4 | 1 | 1 | 128 | 4 |
| $a \times b$ | 0.5 | 512 | 1e-4 | 1 | 1 | 128 | 4 |
| $a/b$ | 0.3 | 512 | 1e-4 | 1 | 1 | 128 | 4 |
| $a^2 + b^2$ | 0.5 | 512 | 1e-4 | 1 | 1 | 128 | 4 |
| $a^2 + ab + b^2$ | 0.9 | 512 | 1e-4 | 1 | 1 | 128 | 4 |
| $a^2 + ab + b^2 + a$ | 0.9 | 512 | 1e-4 | 1 | 1 | 128 | 4 |
| $a^3 + ab$ | 0.9 | 512 | 1e-4 | 1 | 2 | 128 | 4 |
| Needle-in-a-haystack | 0.9 | 256 | 1e-4 | 1 | 2 | 256 | 4 |

## A.3 Additional Results

In Figures 5-11, we illustrate the training and validation accuracy for the arithmetic tasks from above.

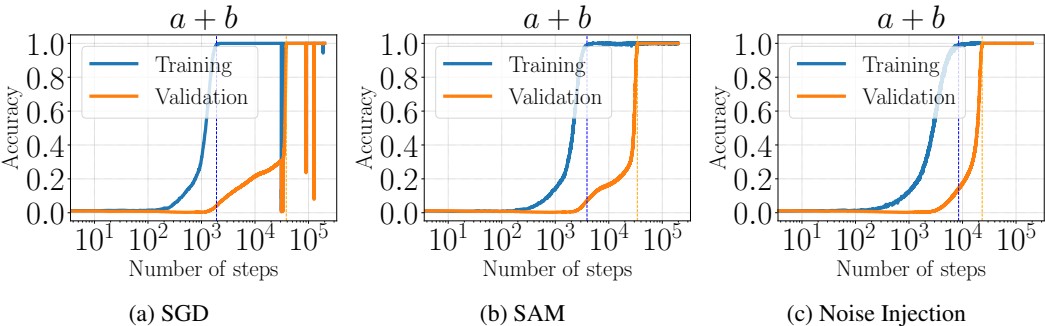

(a) SGD    (b) SAM    (c) Noise Injection

Figure 5: Comparison of SGD, Grokfast, SAM, and our noise-injection method in $a + b \ (mod \ p) = c$.

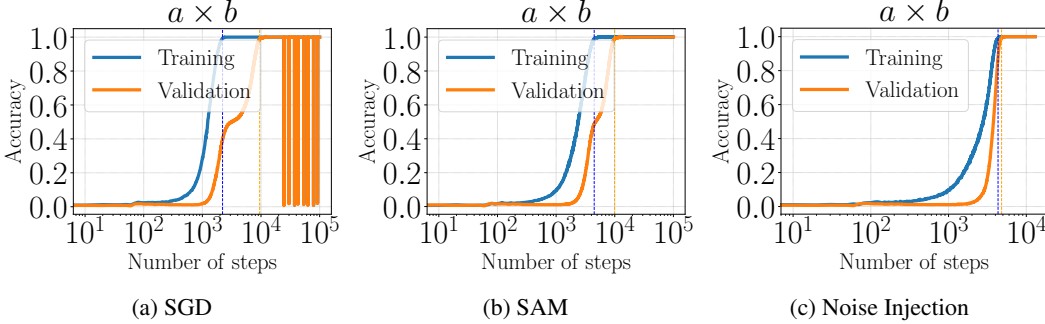

(a) SGD    (b) SAM    (c) Noise Injection

Figure 6: Comparison of SGD, Grokfast, SAM, and our noise-injection method in $a \times b \ (mod \ p) = c$.

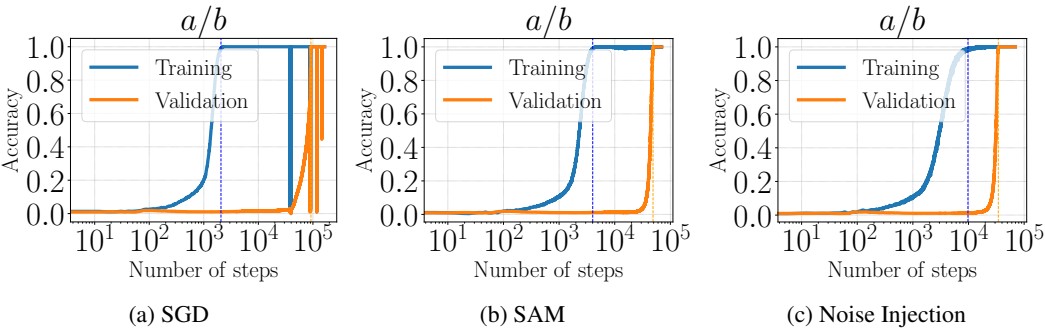

(a) SGD      (b) SAM      (c) Noise Injection

Figure 7: Comparison of SGD, Grokfast, SAM, and our noise-injection method in $a/b \ (mod \ p) = c$.

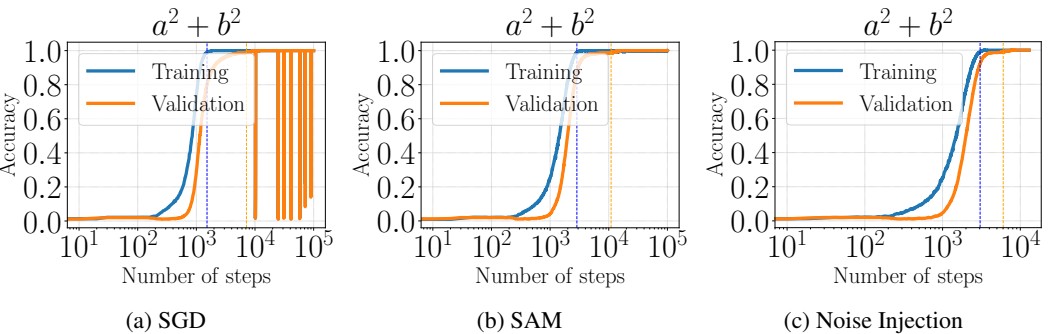

(a) SGD      (b) SAM      (c) Noise Injection

Figure 8: Comparison of SGD, Grokfast, SAM, and our method in $a^2 + b^2 \ (mod \ p) = c$.

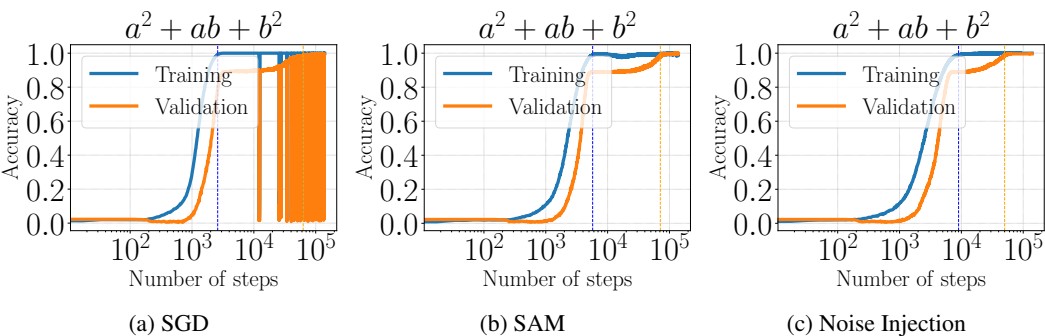

(a) SGD      (b) SAM      (c) Noise Injection

Figure 9: Comparison of SGD, Grokfast, SAM, and our method in $a^2 + ab + b^2 \ (mod \ p) = c$.

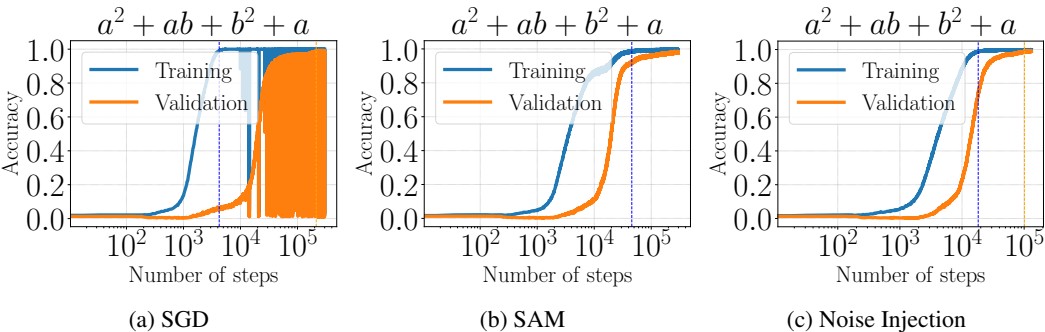

(a) SGD      (b) SAM      (c) Noise Injection

Figure 10: Comparison of SGD, Grokfast, SAM, and our method in $a^2 + ab + b^2 + a \ (mod \ p) = c$.

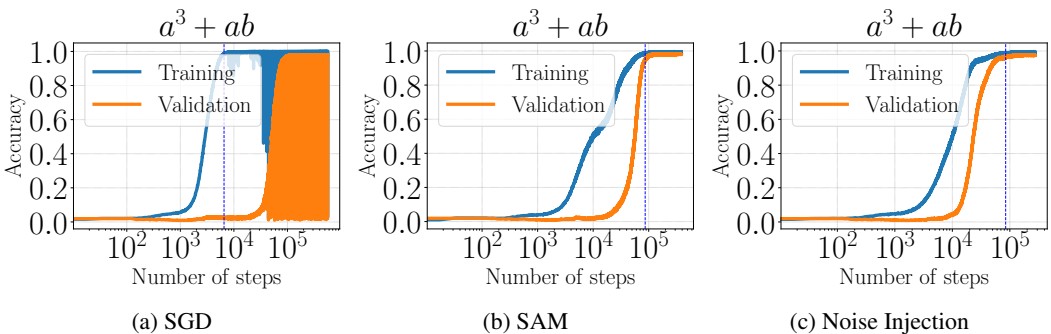

Figure 11: Comparison of SGD, Grokfast, SAM, and our method in $a^3 + ab \, (mod \, p) = c$.

# B  Technical Lemmas

First, we state the result of Taylor's expansion of the perturbed loss.

**Lemma B.1.** *For any $i = 1, 2, \cdots, L$, let $U_i \in \mathbb{R}^{d_i d_{i-1}}$ be a random vector sampled from a Gaussian distribution with mean zero and variance $\Sigma_i$. Let the posterior distribution $\mathcal{Q}$ be centered at $W_i$ and perturbed with an appropriately reshaped $U_i$ at every layer. Then, there exists a fixed value $C_1 > 0$ that does not grow with $n$, such that the following holds for any $x \in \mathcal{X}$ and $y \in \{1, \ldots, k\}$:*

$$\ell_{\mathcal{Q}}(f_W(x), y) - \ell(f_W(x), y) \leq \sum_{i=1}^{L} \left( \langle \Sigma_i, \nabla_i^2 [\ell(f_W(x), y)] \rangle + C_1 \|\Sigma_i\|_F^{3/2} \right). \tag{6}$$

Next, we state the PAC-Bayes bound, which can be found in the PAC-Bayes literature (e.g., [16, 2]).

**Lemma B.2.** *Suppose the loss function $\ell(x, y)$ lies in a bounded range $[0, C]$ given any $x \in \mathcal{X}$ with label $y$. For any $\beta \in (0, 1)$ and $\delta \in (0, 1]$, with probability at least $1 - \delta$, the following holds*

$$L_{\mathcal{Q}}(f_W) \leq \frac{1}{\beta} \hat{L}_{\mathcal{Q}}(f_W) + \frac{C\left(KL(\mathcal{Q}\|\mathcal{P}) + \log \frac{1}{\delta}\right)}{2\beta(1 - \beta)n}. \tag{7}$$

