# OpenReview forum: "A Hessian View of Grokking in Mathematical Reasoning"
_NeurIPS.cc/2024/Workshop/MATH-AI — MATH-AI 24_

### Official Review · Reviewer_Xbeg · 2024-10-06
**The paper is well-aligned with the workshop's theme and clearly presents its contributions, but could benefit from broader analysis across arithmetic tasks and more clarity on computational efficiency and methodology details.**

**Rating:** 6
**Confidence:** 4

**Review:**

**Strength**

_Alignment with the workshop_

The paper aligns really well with the theme of the workshop. The paper proposes regularization of loss surface hessian (equation 1, shows the weight update) to speed up grokking in arithmetic tasks and training stability (reduce variation in training accuracy in the later part of training)

_Writing Style_
* The paper is written well and conveys the core idea without confusion
* The experimental results are presented in a straightforward manner, highlighting the effectiveness of noise injection in stabilizing training.
* The conclusions effectively tie together theoretical insights with empirical observations, offering valuable directions for future research on grokking.


---

**Weakness**


_Limited analysis_

* Though the proposed approach is very effective in increasing grokking speed, the paper only analyzes and shows results on 3 different arithmetic equations. Therefore it is not clear if the approach can generalize for all arithmetic equations.
* Also the paper does not analyze the computation efficiency, the proposed approach requires computing gradients twice with noice added and subtracted from weights (so two forward pass and two backward passes are needed)

_Clarity_
* Equation 1 shows the weight update with loss surface hessian regularization, but the equation doesnt show any weight decay, therefor it is not very clear whether weight decay is used for results shown in Table1.

---

### Official Review · Reviewer_2JdZ · 2024-10-07
**Insightful study but weak relation to reasoning**

**Rating:** 6
**Confidence:** 3

**Review:**

This paper presents a Hessian regularization, which is shown to smooth out the oscillations and accelerate the grokking stage. The authors also find a Hessian-based measurement to estimate the generalization errors.

The idea of making use of Hessian’s ability to capture geometric properties of the loss surface is insightful. Starting from this, the proposals of the new regularization and the new measurement are both novel.
The experimental setup, following some existing work, is also confirmatory. With the hyperparameters clearly listed in the appendix, the discussion around the experimental results is convincing.

One concern is that the paper focuses on modulo arithmetic tasks only, and the generalizability to other mathematical reasoning, the focus of the workshop, is unclear ("analyzing complex information, identifying patterns and relationships, and drawing logical conclusions from evidence").

Overall: insightful study but weak relation to reasoning

---

### Official Review · Reviewer_Yngz · 2024-10-07
**The novelty and contribution of this work is limited**

**Rating:** 4
**Confidence:** 3

**Review:**

This paper investigates the grokking behavior that occurs in arithmetic tasks. The authors propose to address this issue by injecting Gaussian noise into the transformer network, which helps accelerate the grokking stage.

Strengths:
The paper attempts to address the grokking phenomenon which is a common behavior in training, so the technique in this work might be applicable in other settings as well.

Weaknesses:
My concerns regarding this paper are its novelty and its significance. 1) The noise injection technique during the training phase of Deep Learning models is well-known, so the novelty of this work is lacking and its contribution is limited. 2) The technique in this work addresses the training phase but does not really improve the performance of the model. And so, its results are not really interesting. It would be more exciting if the work proposes a method to improve the arithmetic capabilities of LLM, which is still a problem for popular LLMs such as GPT-4o or o1.

---

### Decision · Program_Chairs · 2024-10-09

**Decision:**

Accept

**Comment:**

The ideas presented in the work are interesting. However, as the reviewers point out, the empirical evidence is a little bit lacking and would benefit from more extensive experiments. We suggest that the authors consider including additional experiments to further validate their claims in the camera-ready version.